# Does a mother's childbirth experience influence her perceptions of her baby's behaviour? A qualitative interview study

**Carmen Power**[1]*, **Claire Williams**[2,3]*, **Amy Brown**[1]

1 School of Health and Social Care, Faculty of Medicine, Health and Life Science, Swansea University, Swansea, United Kingdom, 2 School of Psychology, Faculty of Medicine, Health and Life Science, Swansea University, Swansea, United Kingdom, 3 Elysium Neurological Services, Elysium Healthcare, The Avalon Centre, Swindon, United Kingdom

* claire.williams@swansea.ac.uk (CW); carmenpower7@gmail.com (CP)

## Abstract

### Background

Childbirth has become increasingly medicalised, which may impact on the mother's birth experience and her newborn infant's physiology and behaviour. Although associations have been found between a mother's subjective birth experience and her baby's temperament, there is limited qualitative evidence around how and why this may occur.

### Objectives

This qualitative study aimed to explore mothers' childbirth and postnatal experiences, perceptions of their baby's early behavioural style, and whether they saw these as related.

### Methods

A qualitative semi-structured interview schedule collected rich in-depth data. Twenty-two healthy mothers over 18 years of age and with healthy infants aged 0–12 months born at term, were recruited from Southwest regions of England and Wales. Thematic analysis was performed on the data.

### Results

Mothers experienced childbirth as a momentous physical and psychological process. However, they did not necessarily perceive the birth as affecting their baby's early behaviour or temperament. While some mothers drew a direct relationship, such as linking a straightforward birth to a calm infant, others did not make an explicit connection, especially those who experienced a challenging birth and postnatal period. Nevertheless, mothers who had a difficult or medicalised birth sometimes reported unsettled infant behaviour. It is possible that mothers who feel anxious or depressed after a challenging birth, or those without a good support network, may simply perceive their infant as more unsettled. Equally, mothers who have been well-supported and experienced an easier birth could be more likely to perceive their baby as easier to care for.

**Data Availability Statement:** All relevant data are within the paper, including excerpts and the interview schedule.

**Funding:** The authors received no specific funding for this work.

**Competing interests:** The authors have declared that no competing interests exist.

## Conclusions

Childbirth is a physical and psychological event that may affect mother-infant wellbeing and influence maternal perceptions of early infant temperament. The present findings add to prior evidence, reinforcing the importance of providing good physical and emotional support during and after childbirth to encourage positive mother-infant outcomes.

## Introduction

Interventions during childbirth can be life saving for mother and baby. However, some interventions entail risks, where the potential adverse impacts of childbirth complications and interventions on mother and infant health and wellbeing are well-established [1–3]. Certain obstetric interventions may contribute to more unsettled early infant behaviour [4]. Assisted birth in particular, is associated with temporarily disturbed infant physiology and behaviour, including increased crying and cortisol levels [5–7], although not all research has found significant relationships between obstetric complications and early infant temperament [8]. Well-documented associations also exist between pharmacological pain relief (e.g., pethidine/ bupivacaine), newborn behaviour and breastfeeding difficulties [9, 10].

Alongside these physiological impacts of childbirth, the mother's psychological experience may have a lasting impact. For instance, intrusive procedures such as emergency caesarean section (CS) increase the risk of her developing postpartum depression (PPD) after the birth [11]. Furthermore, unfulfilled expectations, fear, or loss of control during childbirth can adversely affect postnatal maternal mental wellbeing, sometimes leading to symptoms or development of post-traumatic stress disorder (PTSD) after childbirth [12–14]. Obstetric complications and interventions together with poor maternal mental health have the potential to disturb early mother-infant relations [15] and breastfeeding [16]. This might be further exacerbated if mothers cannot hold their baby immediately [17], which may disrupt mother-infant physiology, normal perinatal hormone secretion, and instinctive bonding and feeding processes [18–21]. Lack of time and space to bond post birth inevitably heightens negative postnatal mood in the mother, which can impact on the infant's future development [22–24].

Despite these potential associations of the mother's childbirth experience and her infant's behaviour via maternal mood disorders, little is known about the possible effects of subjective maternal childbirth experience on longer term infant temperament development, perceived or otherwise. Although no direct evidence was found, Taylor and colleagues [7] suggested that a mother's response to the birth might act as a 'mediating mechanism' to her infant's own stress response. Experiencing childbirth as traumatic can adversely impact on postnatal maternal psychological wellbeing [25, 26]; and evidence also shows that negative maternal mood significantly influences infant emotionality and behaviour [27–29].

Therefore, the birth experience may affect infant behaviour in the longer term. Indeed, our recent quantitative study of approximately a thousand mothers [30] found that the mother's subjective birth experience and her postnatal mood were associated, and that together they contributed to self-reported infant behavioural style, commonly known as temperament [31, 32]. Consequently, the aim of the present study was to study this in further depth by qualitatively exploring mothers' subjective childbirth experiences. This included the style of care and support received during and after childbirth, mothers' perceptions and interpretations of their baby's behaviour, as well as whether they believed that the two (childbirth experience and infant behaviour) were in any way related.

## Methods and measures

### Design

A qualitative semi-structured interview study [33].

### Participants

Participants were mothers recruited from Southwest England and Wales. Mothers were eligible to participate if they over 18 years of age, were in good physical and mental health, and had a single infant under 12 months. Exclusion criteria were major physical or mental maternal illness, unwell infants, multiple/ preterm/ low birth-weight infants [34], and infants with five-minute Apgar scores of < 7, in accordance with the Welsh Government Maternity Indicator of 'healthy births' [35]. A Department of Psychology Ethics Committee granted ethical approval. Written informed consent was obtained in accordance with the 1964 Declaration of Helsinki. Research questions were developed with care towards the physical and psychological welfare of the participants [36, 37]. Interview length was variable depending on how much mothers chose to say, but typically lasted between 15–60 minutes.

### Measures

A short semi-structured interview schedule (Table 1) using open-ended questioning allowed the free flow of thoughts, feelings, and beliefs without constraint or leading participants [38, 39]. Mothers were asked to tell their birth story, followed by two research questions that examined: (a) their baby's behaviour since birth, and (b) whether they believed the birth had affected their baby's behaviour. The questions were ordered in this way to prevent leading participants. Prompts were used to encourage participants to develop aspects of their birth story and to further explore any new concepts that emerged.

Socio-demographic information was also collected, including age, highest level of educational qualification, occupation, intention to return to work, marital status, and number of children.

### Procedure

A convenience sample of mothers from Southwest England and Wales were recruited via adverts placed in health centres, mother and baby clinics, breastfeeding support groups and social media. Purposive sampling was later employed to recruit both younger and formula feeding mothers. Data were collected between January and October 2015. Data collection stopped when it was judged that no new and meaningful concepts were being 'generated' from the data [40]. Thus, no new themes or subthemes arose during the last few interviews.

Face-to-face interviews were considered the most effective way to capture maternal accounts of their birth stories and perceptions of their baby's behaviour [41]. The interviews

**Table 1. Semi-structured interview schedule for mothers with example prompts.**

| |
|---|
| **Question 1. Tell me about your birth**... |
| • If you were induced, *how* were you induced and how did that feel? |
| • Did you have any pain relief? If so, how did you find that experience? |
| **Question 2. How have you found your baby's behaviour since birth?** |
| • How did your baby seem post birth? |
| • How are they now? |
| **Question 3. Do you think your baby's behaviour was affected by the birth?** |
| • If so, how do you think it affected them? |
| • In what way? |

were mostly conducted face-to-face in participants' homes or nearby cafes chosen by them. Apart from refreshments, participants were not compensated or reimbursed for their time. One interview was conducted via telephone for the mother's convenience. In each instance, the topic was briefly introduced, and participants informed that this was an exploratory study, therefore there were no preconceptions about findings. Informed consent was obtained from all participants, including permission to record the interview by Dictaphone.

No research hypothesis was disclosed, and the researcher's impartiality was emphasised [38]. Afterwards, participants were thanked for their time and debriefed, ensuring no negative effects of being interviewed [38]. Mothers who reported a traumatic birth were encouraged to contact their health visitor or GP and were offered contact details for the Birth Trauma Association.

## Data analysis

Interviews were transcribed verbatim in full and personal identifying information was removed prior to analysis. Participant responses were analysed thematically following the six steps set out by Braun and Clarke [42]. The thematic analysis explored maternal interpretations of events and the meanings they gave to them [43]. Each phenomenon was examined and described as it arose, with the language used by participants and researcher, in accordance with Sandelowski's [44, 45] qualitative description methods.

A thematic analysis was carried out to analyse and arrange the data systematically into themes and subthemes. Credibility of themes and inter-rater confirmability of coding were established through consultation with a second coder (AB). Lincoln and Guba's criteria of 'Trustworthiness' [46, 47] were adhered to as closely as possible. Credibility was established in the approach to recording data and by continuing research until similar experiences and patterns began to appear. Rigorous data coding and analysis processes were performed until data saturation was reached [40], confirming the findings across different mothers. Therefore, the findings might be transferable to similar mother-infant dyads in similar contexts. Nevertheless, it is important to remember that each experience of childbirth is different and largely subjective, as are mother's perceptions of infant temperament. The results were considered dependable and trustworthy due to the involvement of the second analytic coder.

Whilst this study is based in qualitative methodology, it was also considered interesting to note the proportion of mothers who believed their infant's behaviour was related to the birth. Quantifying qualitative data is performed by researchers wishing to further explore data derived from open-ended questioning [48, 49]. However, this was approached with caution and awareness of the limitations attached to quantifying qualitative data in a small sample. Therefore, quantifying aspects of the data did not assume any statistical significance.

**Reflexivity.** The study was designed based on an initial interest in the topic by the first author after her own birth experiences and was further developed after reading the research literature.

The study was conducted with the awareness that past experiences can influence one's interpretation of events, and thus, with a decision to approach the study and the data with an open mind, to welcome mothers' own interpretations of their experiences, and to closely adhere to the data in the reporting [44, 45]. To aid this process, open questions were used in the interviews to allow mothers to speak as they wished about their birth experiences, the data included a wide variety of birth experiences, and there was a second coder. Nonetheless, according to Braun and Clarke [40], there is always an element of subjective interpretation in any qualitative analysis. With an awareness of this possibility, the authors also searched for disconfirming data, and there are examples of this throughout the results section.

## Results

Initially, 23 mothers took part, though one was subsequently excluded as she had previously experienced a stillbirth and was visibly distressed during the interview. Therefore, the final sample consisted of 22 mothers (14 primiparous, mean age = 32, SD = 4.37, Range 21–37). Sociodemographic information is presented in Table 2. As can be seen, the sample covered a good range of mothers recruited from Southwest England and Wales. However, it lacked ethnic diversity as all participants were white Caucasian. The infants had minimum 5-minute Apgar scores of 7 and a mean age of 22.5 weeks (SD = 13.16, Range 1.3–41.7 weeks).

### Data summary

Of the 22 mothers who met all the inclusion criteria, there were three planned inductions, two planned caesareans, and eight unplanned inductions or accelerations of labour. Nine women had spontaneous physiological births, seven of which were waterbirths. With the exception of planned interventions, mothers appeared to be more likely to report their baby's behaviour as unsettled after a difficult or medicalised birth (n = 6), and more likely to report calm, settled newborn behaviour after a straightforward birth (including a planned induction or caesarean) without unexpected complications (n = 10). However, less than half the mothers (n = 10) perceived the birth as affecting their baby's behaviour, and several (n = 4) felt ambiguous.

### Thematic analysis of mothers' birth stories and their beliefs and perceptions on how this may or may not have affected their baby's behaviour

Nine childbirth themes were identified relating to mothers' perceptions of how they felt they were affected by objective/physical birth events (e.g., type of birth) and subjective/ psychological factors (e.g., feeling empowered, well cared for and supported, frightened or joyous), as well as how these factors may have influenced their baby's behaviour post birth. In addition,

**Table 2. Sample distribution by sociodemographic factors.**

| Indicator | Group | N | % |
|---|---|---|---|
| Age | 20–24 | 2 | 9.09 |
| | 25–29 | 3 | 13.64 |
| | 30–34 | 9 | 40.91 |
| | 35 > | 8 | 36.36 |
| Education | School | 2 | 9.09 |
| | College | 5 | 22.73 |
| | Higher | 15 | 68.18 |
| Maternal occupation | Professional | 10 | 45.45 |
| | Skilled | 8 | 36.36 |
| | Unskilled | 3 | 13.64 |
| | Other (student) | 1 | 4.54 |
| Marital status | Married | 15 | 68.18 |
| | Cohabiting | 7 | 31.82 |
| Number of children | One | 14 | 63.64 |
| | Two | 7 | 31.82 |
| | Three | 1 | 4.54 |

*Note*: *Occupations coded according to the Office for National Statistics (ONS) Socio-economic Classification* [50].

mothers spoke about the postnatal days and bonding with their baby, which generated two more themes. Consequently, maternal birth story data are presented in three categories with corresponding themes and subthemes: (1) Physical birth experience, (2) Psychological birth experience, and (3) Postnatal days with baby (Table 3).

The themes and subthemes are presented in full below. Each theme is represented by direct maternal quotes, first on how they felt that variable affected her own experience, and then on how some mothers felt it may have affected their baby.

**1. Physical birth experience.** The physical birth experience refers to the mother's reported experience of events, such as what happened during the birth. This included how they felt they were affected by the type of birth, the birth and postnatal environments and the pain relief method. Mothers also commented on how they were affected psychologically throughout their descriptions of these physical birth themes. It is notable that all the mothers who experienced challenges and thus appeared to have more negative perceptions of their birth (n = 8), had undergone a medicalised birth with one or several interventions. Meanwhile, 75% of mothers who had experienced their birth as 'positive' (n = 8) had a spontaneous physiological birth (n = 6). The two remaining women who experienced their birth as positive had planned interventions (one caesarean section and one induction). However, as the sample is too small for these numbers to have any statistical significance, the themes are explored qualitatively.

a) *Perceptions of birth mode.* Many mothers described how the type of birth they had (i.e., caesarean, forceps, spontaneous etc.) affected their perceptions of their birth experience. Spontaneous physiological labour and birth could be experienced as empowering, enabling mothers to feel confident post birth.

*"It sort of burns a bit but it's not pain. . . 'cause you can feel something happening. . . It just happened. I couldn't stop it. . . My birth was incredible! I'm very lucky. . . I got in the pool*

Table 3. **Summary of thematic analysis: Maternal experiences of childbirth and how they felt this might have affected their baby's early behaviour.**

| Category | Name of theme/subtheme | Definition |
|---|---|---|
| **(1) Physical birth experience** | (a) Perceptions of birth mode | How the type of birth was perceived to affect mothers and infants |
| | *i) Spontaneous* | Physiological labour and birth |
| | *ii) Obstetric interventions* | Interventions during childbirth e.g. induction, acceleration, forceps |
| | *iii) Operative birth* | Caesarean section |
| | (b) Qualities of the birthing environment | How the birthing environment was felt to affect mothers and infants |
| | *i) Home* | Homebirth–physiological birth only |
| | *ii) Midwife led unit (MLU)* | MLU–stand-alone/ adjacent to hospital |
| | *i) Hospital* | Hospital birth–choice or complications |
| | (c) Perceived effects of pain relief | How types of pain relief were felt to affect mothers and infants |
| | *i) Pharmacological* | Medical |
| | *ii) Natural methods* | Non-medical |
| **(2) Psychological birth experience** | (a) Emotional states | How mother felt during labour and birth |
| | *i) Joy* | Feelings of joy and wellbeing |
| | *ii) Fear* | Feeling unsafe and afraid |
| | (b) Expectations | Maternal expectations of childbirth |
| | (c) Health Professional Authority | Midwife/doctor stipulating or refusing a treatment or intervention |
| | (d) Support | Social support–from birth partner/ healthcare professional |
| | (e) Neglect | Personal needs were unmet |
| | (f) Separation | From partner or baby or both |
| **(3) Postnatal days with baby** | (a) Baby mirrors mother | The baby seeming to reflect its mother's emotional state |
| | (b) Maternal caregiving | The mother's style of caring for her baby post birth |

*and little K was born fifteen minutes later."* (Mother 12, home waterbirth; accidental freebirth)

Conversely, obstetric interventions could lead to distress in some instances, leaving the mother feeling helpless or in increased pain.

*"I didn't progress at all. . . then they gave me the top up (synthetic oxytocin). . . it just over-stimulated my uterus. . . absolute agony."* (Mother 22, induction, acceleration, pethidine, Entonox, spinal block, neonatal distress, Neonatal Intensive Care Unit)

Despite neonatal distress and her baby's admission to the Neonatal Intensive Care Unit [NICU], this mother did not believe there was any connection between the birth experience and her baby's very active temperament. Although she reportedly had feeding problems and sickness for the first few weeks, the mother said that her baby soon settled into easier feeding and sleeping routines.

*"She's a really contented happy baby. . . She's really active. . . She wants to do everything, she wants to be in everything. She's just so content, she just plays by herself. . . I don't think it's anything to do with that cos while I was pregnant she was quite a busy bod then. I think it was just in her nature."."* (Mother 22, induction, acceleration, pethidine, Entonox, spinal block, neonatal distress, NICU)

One mother who experienced an operative birth said it felt easier at first but more challenging during recovery.

*"I had a really easy and pleasant few hours with her whereas a lot of mums are taken off for stitches or in another room, or in lots of pain. I didn't have that, but then I think the day after when most people start to feel a lot better, I felt so much worse."* (Mother 4, planned CS, postpartum: internal haemorrhage, morphine)

Nevertheless, this mother felt that her baby had benefitted through having a calm caesarean:

*"Yes. There was a calm mummy and a calm daddy and a happy atmosphere. No pain, no upset, cuddles straight away, music in background. . . She didn't have any of the trauma of birth and just immediately she had all my love and attention. . . and all Ed's love and attention. And even when I was in pain the next day she got so many cuddles from Ed and his parents. . . So her character is calm, happy and interested."* (Mother 4, planned CS, internal haemorrhage)

Mothers who had planned interventions (such as a planned induction or a planned caesarean) seemed more likely to view their birth more positively overall than those who had unplanned interventions, although this was not always true.

*"I didn't feel any pain, but I feel like I wasn't prepared enough for how much I might feel, 'cause you can feel each layer dragging back. . . horrible. . ."* (Mother 13, planned CS, postpartum infection)

Although this mother had a difficult birth and postnatal experience, and her baby had trouble breastfeeding initially, she did not feel that the birth had affected her baby.

*"I couldn't breastfeed which I wanted to do but he wasn't latching on and I was in agony any-way, so I was expressing. . . I don't know. He stayed sort of calm like when he was born so I don't think so really—obviously things could have gone wrong but they didn't for him. I was worried the antibiotics would affect him but they didn't. No thrush. . ."* (Mother 13, planned CS, postpartum infection)

*b) Qualities of the birthing environment.* The birth environment could also affect mothers' per-ceptions. Notably however, although some experienced fear and lack of control in hospital, many found hospital reassuring.

*"I really felt I needed just to know that people were with me if anything happened. . . It was brilliant."* (Mother 11, physiological hospital birth)

Mothers who actively chose to birth in hospital felt much happier about the birthplace than those who had not actively participated in this choice, but instead felt they had succumbed to pressure during labour. Such hospital transfers could be experienced as very distressing.

*"I had planned a pool birth in the midwife led (unit) but when I went back they sent me up a floor (to the hospital) . . . I was absolutely devastated that I couldn't go down. . . I'd laminated me birth plan!"* (Mother 14, acceleration, episiotomy, forceps)

Several mothers who birthed in a midwife led unit praised the high quality of care, finding it *"homely. . . more natural".* (Mother 10, MLU waterbirth)

*"It was a fantastic experience really. . . The midwife care was brilliant and the facilities there are great. . . So the midwife caught her and then just passed her over to me (laughs) and then S cut the cord."* (Mother 20, MLU, waterbirth, Entonox, tear)

This mother believed the positive birth experience may have benefitted her baby's temperament.

*"Pretty much everything that's around her that has helped her to feel safe and secure and calm. And yeah, maybe that started off with having a well and truly sort of straightforward arrival."* (Mother 20, MLU, waterbirth, Entonox, tear)

The homebirths in this sample (n = 2) were both waterbirths which involved little pharma-cological pain relief and no obstetric interventions. They were perceived as calm and positive experiences, despite the midwife not arriving in time for one of them.

*"Yeah, it felt very calm in the house, really calm. . . I was in bed until pretty much he was com-ing down. . . I was very relaxed. . . it wasn't even too painful."* (Mother 12, home waterbirth; accidental freebirth)

This mother felt that being relaxed herself during the birth may have contributed to her baby also being relaxed.

*"It was a very calm birth, and he was pretty calm coming out. . . he was very settled. . . I think that a waterbirth is definitely a relaxing way, definitely relaxing for me, so it must have some impact on them."* (Mother 12, home waterbirth; accidental freebirth)

c) *Perceived effects of pain relief.* The type of pain relief used during labour, or a lack of pain relief when requested, could affect the overall birth experience. Participants reported variable experiences. Although it did not agree with everyone, many enjoyed the mildly analgesic effects of Entonox (gas and air).

*"I loved it (Entonox), it was fantastic after nine months of sobriety (laughs), it was really great."* (Mother 20, MLU, waterbirth, Entonox, tear)

Some mothers who had not initially wanted any pain relief were relieved to have an epidural after hours of 'agony'.

*"Once they'd put the epidural in my back, the rest of the experience was just amazing. . . for 8 hours I just sat there. . . and really enjoyed it. Then they said, 'I think you should push,' and I pushed for ten minutes. . ."* (Mother 6, hospital, acceleration, epidural)

This mother did not find her baby affected by any aspect of her birth experience, including the epidural.

*"I don't think so, but I don't know. She had a pretty easy coming into the world. Whether my stress hormones in the earlier part of labour got through I don't know. . . but she didn't get distressed, no." (*Mother 6, hospital transfer, acceleration, epidural)

Experiences of pethidine in this sample were either neutral or negative.

*"It just made me sleepy, really really sleepy. I can't say that it really took away much pain, but it just made me just completely relaxed. . ."* (Mother 18, induction, pethidine, episiotomy, cervical tear, postpartum haemorrhage)

*"I didn't really think much of that (pethidine) because it just made me feel like I wasn't there, had no control, but I still had pain."* (Mother 5, hospital waterbirth, pethidine)

One mother described contractions under the influence of pethidine as a sudden shock.

*"I think I went into transition phase whilst I was asleep. I woke up and just screamed like blue murder. . . you know, like one of those really like. . . blood curdling screams. . ."* (Mother 10, MLU waterbirth, pethidine, Entonox, NICU)

Two mothers thought their baby's early behaviour could have been affected by the pethidine.

"No, the only thing that I was worried about in the beginning was she slept loads and loads and loads and I was worried that it was something to do with the pethidine. She slept like a

ridiculous amount in the beginning and they had to say like, 'Have you fed your baby? Wake her up!'" (Mother 5, hospital waterbirth, pethidine, Entonox)

*"She's definitely a more clingy baby than my first. She does not like to be put down, she doesn't lie down, she's quite sicky. . . She won't settle away from me. My mum thinks it's all the chemicals she had."* (Mother 15, induction, forceps, pethidine, epidural; NICU: infant infection, antibiotics)

Alternative techniques and non-medical forms of pain relief sometimes helped mothers to cope without side effects. Methods of relaxation such as hypnobirthing were not always perceived as providing the relief originally envisaged.

*"The only thing I really took away from the hypnobirthing was the breathing. The breathing got me through."* (Mother 6, hospital transfer, acceleration, epidural)

Waterbirths were generally found to be relaxing, although this feeling did not always last.

*"The first contractions in the pool were so blissful, then the novelty wore off. . . very quickly I needed to push."* (Mother 3, home waterbirth)

Mother 3 went on to describe her baby's behaviour as 'easy'.

*"Oh, he's been so calm and content and he's been very easy. . . He is generally very happy, smiles a lot, is very good at being passed around. . . very calm.* (Mother 3)

**2. Psychological birth experience.** As can be seen from the descriptions above, the mothers' psychological birth and postpartum experiences were quite connected to and interwoven with their physical birth experience. The subjective psychological experience often dominated birth stories.

*a) Emotional states during childbirth.* Emotional states during the birth were important to mothers and stayed with them afterwards. Some mothers appeared to have actively enjoyed giving birth.

*"It was very much my body did it, and like I enjoyed feeling it."* (Mother 11, physiological hospital birth)

*"I didn't feel any pain, so I was just going with it. I was enjoying it, in the moment. . . All this bit I remember like it was the best thing I've ever done. It was amazing like. The whole like established labour and actually getting him out was amazing."* (Mother 10, MLU waterbirth, pethidine, Entonox, NICU)

However, lack of labour progression could lead to fear, especially if this meant medical aid in the form of unexpected interventions.

*"So maximum dosage (of synthetic oxytocin) and the surgeon was ready behind the curtain– they were polishing their cutlery. . ."* (Mother 23, acceleration, hospital birth)

This mother was not sure whether her emotional state during the birth had affected her baby.

*"Yeah, I think in the early days, I do think umm. . .(pause) it's really hard to say. I don't know whether she was affected, or I was affected. . . It was just a bit on edge that birth. . . just a bit on edge."* (Mother 23, acceleration, hospital birth)

One mother claimed to feel calm once labour commenced, despite her initial fear and anxiety.

*"I phoned my mum and I said, 'I'm so scared I feel really calm.' You know when they tell you to write a birth plan, they tell you to prepare yourself, pack your bags, everything. . . I suddenly felt like everything was out of my control and it almost relaxed me completely. . . because I no longer could control it, I think, if that makes any sense."* (Mother 11, physiological hospital birth)

*b) Expectations.* Maternal expectations of the birth and postnatal period was an important theme. Mothers appeared to be more accepting and viewed their birth experience in a more positive light when they felt mentally prepared (e.g., for a planned induction or planned caesarean section).

*"Well, I had to be induced because of my diabetes so we went into hospital on the Saturday. . . A little cut and the head popped out. It was lovely."* (Mother 19, planned induction)

High expectations of a positive birth experience could lead to acute disappointment post birth.

*". . . because I used to be super fit and healthy and run half marathons and I thought, yeah, I can do this. I thought I'm the sort of person who's got a body that can deliver a child. . . Wrong way round!"* (Mother 4, emergency CS)

Mothers who were unexpectedly kept in hospital post birth because they or their baby needed medical treatment could also find this very difficult.

*"That was the worst week, well five days, of my life. It really traumatised me, that stay in hospital. . . which is really hard because when you speak to some people they say, 'Oh we loved it, you know, the support we got from the midwives. . .' I said quite often, 'This is worse than prison. In prison you get an Xbox, and you get a TV, and you don't have a baby that's waking you up every half an hour.'"* (Mother 10, MLU waterbirth, Entonox, pethidine, NICU)

Although this mother had a waterbirth, she felt the long labour may have affected her baby's wellbeing and behaviour alongside his medical issues and five days in NICU post birth.

*"I think so. . . Yeah. . . I think his difficulties feeding were affected by the birth. . . the length of it. . . I felt like he would have suffered for quite a long time. . .being squashed and stuff, because*

*it was going on for so long. . . Yeah, I can't really put my finger on why it would have affected him but I'm sure it would. . ."* (Mother 10)

Mothers who felt they had experienced an unpredictably difficult birth which involved feelings of trauma, could suffer conflicting emotions postnatally.

*"And I think as a result of that as well I didn't feel I could bond with him at all well. . . We really struggled with him the first couple of weeks. 'Have I done the right thing having a baby?' All this jazz."* (Mother 14, hospital transfer, acceleration, episiotomy, forceps)

This mother attributed her baby's difficulties breastfeeding to the forceps, pain and distress.

*"I put it (breastfeeding problems) down to the fact that, because he was forceps. . . when I was trying to put him on the boob, I was pressing the back of his head and neck, not realising. . . So I felt really upset that I was really useless and couldn't do it."* (Mother 14, hospital transfer, acceleration, episiotomy, forceps)

In contrast, mothers appeared to fare better postnatally if their prenatal expectations of childbirth had been met.

*"I found both births really empowering. . . quite euphoric. They gave me a sense of achievement."* (Mother 7, physiological hospital birth)

Although this mother did not connect her baby's contented behavioural style to the birth.

*"Sleep was good. He'd sleep for 5–6 hours at a time. . . he's quite a content little baby. . .as long as he's fed he will self soothe, let me leave the room, go to anybody. . . No, I don't think so. . . I think cos I didn't have a traumatic birth. . .but I think if I had experienced trauma it may have affected him."* (Mother 7, physiological hospital birth)

c) *Health professional authority*. Several women reported agreeing to interventions for their baby's safety without feeling they were given sufficient information, time or opportunity to make an informed decision.

*"And then everything I didn't want to happen happened which was, I didn't want to have stirrups or anything like that, and the next thing was they took the gas and air away and said I wasn't pushing hard enough. . ."* (Mother 2, hospital induction, third degree tear, postpartum haemorrhage)

However, this mother saw no connection between a traumatic birth experience and their baby's behaviour in the early postnatal period, despite acknowledging her baby's discomfort.

*"No, not directly I don't think. If he'd been born naturally premature, we may have got a bit more support. . . Looking back there wasn't a time when he wasn't crying. . . He'd feed little and often 'cause he was so uncomfortable as well. . . He wasn't sleeping and nor was I. . ."* (Mother 2, hospital induction, third degree tear, postpartum haemorrhage)

One mother explained how she felt coerced into agreeing to unwanted interventions.

*"If the doctors decide this is the way they want you to do it it's very hard not to. If something then happened. . . I'd never forgive myself."* (Mother 15, hospital induction, forceps)

On occasion, women's privacy was violated, sometimes leading to deep psychological distress.

*"They broke my waters–they didn't ask. Suddenly there was stuff coming out of me. I did not know what was happening . . . They were telling me, 'Keep your legs open, keep your legs open', though my natural instinct was to curl up in foetal position to protect myself."* (Mother 1, overnight hospital induction)

Even standard interventions such as electronic foetal monitoring could be restrictive and uncomfortable, and again, mothers did not always feel fully consulted beforehand or that they had any choice. 'Not allowed' was a common phrase used to describe this lack of control.

*"I wanted to do it on all fours. . . 'cause I was connected to all the machines I wasn't really allowed to move. . ."* (Mother 19, planned hospital induction)

*"And I said, 'Well I'm having a waterbirth' and he (Dr) said, 'Oh no you're not, you've gone over 24 hours now'. . . so I lost the plot and burst out crying. . ."* (Mother 14, hospital transfer, acceleration, epidural, episiotomy, forceps)

*d) Support.* Support during labour and birth as well as postnatally seemed essential to women's overall wellbeing. Mothers who received emotional support felt well cared for, which appeared to positively affect their perceptions of the whole birth experience.

*"She (midwife) was brilliant. . . She really looked after me afterwards. . . She even washed me. . . Yeah, she was very, very special."* (Mother 11, physiological hospital birth)

A well-supported birth experience could positively influence a mother's postnatal mental state and her perceptions of her baby, which this mother believed were associated.

*"He's really happy, he's really placid as well. . . Urm, well yeah and particularly my um emotions I suppose. . . like I felt confident and calm and happy. . .and I still do."* (Mother 11, physiological hospital birth)

Fathers who were able to remain in the birth room throughout labour could also play an important supportive role.

*"My husband was doing all the things he'd learnt. . . supporting me, um rubbing my back, making sure I was drinking. He also did some of the things he wasn't taught to do which was um get lots of kidney dishes because I was vomiting a lot* (laughs).*"* (Mother 16, physiological hospital birth, Entonox)

This mother thought her baby's temperament may have benefitted through the type of birth she had, and felt they were well set up to bond immediately.

*"We didn't really have any drugs, so she came out as alert as she was going to be. It was quite nice to know that we could bond straight away, and she was a hundred per cent rather than dosed up on pethidine or something."* (Mother 16, physiological hospital birth, Entonox)

Health professional postnatal support, especially for breastfeeding, was appreciated by new mothers, particularly if they were finding it difficult to get established.

*"I was lying to the midwife, saying she was latching on 'cause I was so desperate to go home... then this lovely Columbian midwife saved the day. She literally grabbed my boob and flung it in her (baby's) mouth (laughs). So that's how it's done!"* (Mother 6, hospital transfer, acceleration)

*"She's just amazing (health visitor)–I call her the boob whisperer 'cause... She calmed me more than anything... She just chilled me out and I didn't feel like she was judging me because I can't do it."* (Mother 14, hospital transfer, acceleration, forceps)

*e) Neglect*. However, busy maternity wards could lead to intermittent midwifery care. Women who had such experiences could feel abandoned and neglected and become fearful about something happening to them or their baby. One mother ended up giving birth without any pain relief due to lack of staff attendance, though she and her partner had requested it several times.

*"Yeah, we kept getting left a lot in the room on our own with nobody–it was quite scary... The pain was like getting worse and worse. I was getting really dizzy and sick, but... they wouldn't give me any pain relief 'cause they just thought I was really whingey."* (Mother 17, hospital induction, third degree tear)

Despite her frightening experience both during and after labour, this mother saw no connection between the birth and her baby's unsettled behaviour.

"*The first few weeks he was just constantly sick, constantly in pain and screaming.... (he's) jittery. He does it all the time, it's like he's startled and his hands sort of shake.*" (Mother 17)

Neglectful care provoked conflicting emotions, ranging from empathy with overrun staff to personal distress. This could occur in the postnatal ward as well. While mothers tried to be considerate of the busy midwives, at times their own and their baby's needs were left unmet.

*"The antenatal care was good but the postnatal care... I was left in dirty sheets, I had to shower myself and at times I had to sit on the floor in the shower because I couldn't stand up... One night I was left with a drip in one arm but the baby over the other side..."* (Mother 4, planned CS, internal haemorrhage)

*f) Separation*. Women were sometimes separated from their partners during night inductions when the partner was sent home (Note: maternal data for this study were collected pre COVID-19).

*"I was just on my own, honestly, it's horrible... I think your partner should be with you... I was quite lonely... I didn't sleep..."* (Mother 15, hospital induction, forceps)

One woman remarked on the feeling of emptiness when unexpectedly left alone after the birth.

*"Then I was left on my own in that room for 2 hours. It was strange after all that, and all the people, suddenly completely alone."* (Mother 6, hospital transfer, acceleration)

Mothers who were separated from their baby and partner post birth could feel distraught.

*"I was taken. . . to have a shower and he went with his dad to NICU. . . I suddenly panicked–Where's my husband and my baby?"*

(Mother 10, MLU waterbirth, pethidine, Entonox, NICU)

This mother felt that her baby's behaviour may have been disrupted by the stressful circumstances around the birth and the way this affected her.

*"I think so yeah. . . I know it definitely affected me (laughs), and that in turn probably affects him as well—and that I was just so tired—but I can't specifically say what. . . I think his difficulties feeding were affected by the birth. . . and also he was taken away from me straight away. . ."* (Mother 10, MLU waterbirth, pethidine, Entonox, NICU)

Although Mother 10 also acknowledged that it may have been direct postnatal circumstances (i.e. being in NICU) that affected her baby's wellbeing and behaviour at the beginning.

*"He was waking a lot in the night to feed. And then probably from about two months old he's been a really laid-back baby. He's been really calm. . . (In NICU) they were doing obs. every half hour which was very disruptive for him. . . he'd scream and cry. . . He didn't have a very nice start."* (Mother 10, MLU waterbirth, pethidine, Entonox)

**3. Postnatal days with baby.** In addition to their birth stories, mothers spoke frequently about their postnatal time with their baby.

*a) Baby mirrors mother.* Some mothers perceived their baby's behaviour as a response to their own internal state.

*"Part of me wonders if her behaviour's been affected by my reaction to the birth if that makes sense. . . she is very different to the first in terms of her neediness. . ."* (Mother 15, induction, forceps, pethidine, epidural)

Mother 19 believed her baby's post birth behaviour reflected how she herself had felt during childbirth.

*"I suppose having a chilled out relaxed-ish birth might help–I wasn't stressed out at all. It helps the baby not feeling any stresses from the mum. Yeah, he's a very chilled out little boy."* (Mother 19, planned induction, Entonox)

Some spoke about having a traumatic birth and an unsettled infant, although these mothers were often unsure whether there could be a connection between the two.

*"It was traumatic, horrific. . . first they did an internal examination. . . I felt. . . kind of exposed. . . I spent the whole night in absolute agony. . . like someone ripping my insides*

*apart. . . . They broke my waters. . . I didn't know what was happening. . ."* (Mother 1, overnight hospital induction)

Mother 1 also described her baby's behaviour and early motherhood as very challenging, although she did not perceive this to be associated with the birth.

*"Very angry when he's changed or bathed. . . he screams his head off. Feeding. . . was a battle. It feels like the hardest job I've ever had."* (Mother 1, overnight hospital induction)

Mothers who had a spontaneous physiological birth could also at times find it challenging.

*"I don't think I had a traumatic birth, but. . . being told nothing was happening and thinking, I can't cope with it if that's the case. . ."* (Mother 8, MLU waterbirth, no medication)

This mother went on to describe how she experienced her baby's behaviour as 'difficult' due to her own state of mind after the birth.

*"I think I found her quite difficult because I was a little bit shell-shocked after the birth. . . in the early days. . . I mean we did have some days in the early days where she was really hard to console for no reason other than maybe feeling a bit colicky. . ."* (Mother 8, MLU waterbirth, no medication)

*b) Maternal caregiving.* One mother felt her unsettled newborn baby become calmer through keeping them close after a difficult birth experience.

*"I don't know whether she was affected, or I was affected, but I did just cuddle her and molly-coddle her for a long time. . . She was contented on me. . . So yeah, we were quite stuck."* (Mother 23, hospital acceleration, no pain relief, breathing techniques)

Mothers were generally responsive to their baby's needs and willing to do things differently if needed.

*"I co-sleep with this one which I never did with my first, mostly because it's the only way I can get some rest."* (Mother 15, induction, forceps, pethidine, epidural)

This could mean that other family members were sometimes excluded in the early days.

*"I didn't really let others in as much to hold her and carry her round and things when she was a tiny baby 'cause I just felt like she just needed to be back in the womb* (laughs).*"* (Mother 23, hospital acceleration, no pain relief)

## Discussion

This study conducted an in-depth exploration of maternal perceptions of childbirth and their infant's behavioural style post birth. Notably, fewer than half of the mothers perceived their baby's behaviour to have been influenced by their birth experience. After outlining a challenging or traumatic birth, some mothers proceeded to describe their newborn as unsettled, without attributing this in any way to the birth. It is possible that these mothers did not perceive or wish to acknowledge a potential association between the birth and their baby's behaviour because of guilt, disassociation, or denial after a difficult birth experience [51]. Alternatively,

the seeming incongruity between maternal reports of their birth, their baby's behaviour and their beliefs around this, could stem from an external reason, such as difficulties with feeding, lack of postnatal support, or maternal worry about financial security.

For others whose descriptions of their experiences appeared to align with their perception of their newborn baby as relatively 'calm' or unsettled', the relationship, if it did exist, could be multifaceted. For example, mothers who felt positive during pregnancy may have had easier births and more settled infants due to reduced maternal anxiety—benefitting both the birth experience [52] and her baby's behaviour [22, 53]. Possible explanations for the findings are discussed below.

## Physical birth experience

Although many mothers perceived no connection between their birth experience and their baby's behaviour, some who reported an easy, straightforward birth, such as a calm and well-supported waterbirth, connected this experience with their babies seeming content settled. In contrast, others perceived that difficult birth events, such as forceps delivery, might be linked to their baby's unsettled behaviour due to residual pain and distress. The mother's psychological experience of birth may play a role here but there are also potential physiological links. Prior research shows higher levels of umbilical cord cortisol in infants experiencing obstetric complications, especially assisted birth [5, 7, 54].

Moreover, infants are affected by their mother's raised cortisol levels during labour [55]. Maternal cortisol release can alter the development of the infant's HPA axis governing their future cortisol production [56]. This may occur alongside changes to placental and immune system functioning, the infant's genome and epigenome, and gut microbiota. In turn, epigenetic changes and disrupted gut microbiota are associated with excessive crying and fussing in infancy [57] and negative effects on future health and behaviour [58, 59]. Specific risk factors such as induction, assisted birth and general anaesthetic [60], may alter the neonate's future stress responsivity [5, 61] and increase their reactivity and fearfulness [53]. Mothers in our study, however, were more likely to simply perceive their baby's pain or discomfort after a forceps birth rather than the bigger picture of increased stress reactivity.

The birth environment seemed to influence mothers' sense of wellbeing during childbirth. Although many women found being in hospital reassuring, some mothers who experienced an in-labour transfer from home or MLU to hospital reported becoming distressed. In some places, midwifery services appeared to be stretched despite the most recent National Institute for Health and Care Excellence (NICE) recommendations of one-to-one care for all labouring women [62]. Inconsistent care and noticeable staff shortages on the postnatal ward could also be experienced as distressing, negatively impacting on perceived quality of care [63, 64].

Pain relief could affect mothers and infants in complex ways, with some mothers praising the positive benefits of Entonox or epidural, but with others feeling no effective pain relief after pethidine [65]. Two mothers felt that pethidine may have affected their baby's sleepiness, aligning with prior evidence that it can reduce newborn alertness and contribute to breastfeeding difficulties [10, 66, 67]. It may also increase the likelihood of admission to NICU [68], although not all research shows such an effect [69].

A few mothers reported initial breastfeeding problems after combinations of pethidine, synthetic oxytocin and/or epidural, which together may block endogenous oxytocin production: the essential hormone for effective birth and breastfeeding [70]. These findings are consistent with evidence that newborn alertness and suckling behaviours can be disrupted by these combinations of medical pain relief methods [71]. In addition, epidurals increase the risk of assisted birth and infants being admitted to NICU, which may further increase breastfeeding challenges [72].

## Psychological birth experience

The potential pathway between mothers' experiences of childbirth and descriptions of their baby's early temperament might also be explained through the mother's psychological birth experience and subsequent postnatal state. The psychological experience incorporates both physiological and emotional responses to the birth. For example, experiencing fear during childbirth may increase maternal cortisol production, which increases perceptions of pain [73] and passes readily through the placenta and breastmilk, potentially heightening the unborn infant's future fear reactivity [74].

Whether or not they perceived any connection, mothers with more positive birth experiences which went smoothly and where they felt well-supported, listened to and cared for, tended to describe their infants as calmer and more settled than mothers who seemed to have had a less positive experience. Negative childbirth experiences could stem from unmet expectations and how mothers felt they were treated by caregivers during labour, birth or postnatally. 'Mistreatments' of women during childbirth have been defined as: awareness of staff shortages, feeling inadequately supported or neglected, feeling their birth choices were disrespected, not being 'allowed' to bring their partner into the bathroom, lack of informed consent, painful vaginal examinations, and health professionals refusing to provide adequate pain relief [75]. Although these factors go directly against the WHO's recommendations [34] and the NICE guidelines for intrapartum care of healthy mothers and babies [62], they were all reported at some point by mothers in our sample.

Mothers sometimes reported feeling distressed or fearful during unplanned obstetric complications or interventions. These mothers appeared more likely to report experiencing initial difficulties with their infant's feeding or sleeping behaviours, adjusting to motherhood and bonding with their baby. This would align with the theoretical model in our previous paper which proposes a potential pathway from the birth experience to postnatal maternal mood, mother-infant bonding and infant behaviour [30]. In contrast, mothers who had a spontaneous physiological birth seemed more likely to mention feelings of joy, empowerment or elation during and after childbirth, and these mothers also appeared more likely to perceive their baby's temperament as 'easy going' and to bond easily. This fits well with previous findings that spontaneous physiological birth is associated with a release of neuropeptides, prolactin, and oxytocin [18, 19], all of which are known to have a positive impact on maternal mood and mother-infant attachment behaviours [76].

Mothers' expectations of childbirth also seemed to influence their postnatal mood and could be connected to the way they perceived the birth and early motherhood [77–80]. Mothers who had unmet expectations and disappointment around childbirth sometimes reported their newborn as more unsettled and occasionally also found it difficult to bond with their baby. Potentially antenatal education could prepare mothers better for the reality of childbirth and parenting and ensure that all parents know where to go if they need to seek help postnatally.

Perceived 'health professional authority' could occur when women experienced a loss of control. This sometimes happened around hospital transfers or unwanted interventions such as continuous foetal monitoring, which is standard hospital practice despite the lack of evidence to support it. Some participants who were not fully informed or felt 'coerced' into consenting to unplanned interventions had regrets around their birth, potentially impacting their early experiences of motherhood and the way they perceived their newborn. 'Lack of control' has been cited as the most common factor in traumatic childbirth (54.6%), seconded by intense pain (47.4%) [81]. A Swiss survey of 6054 women also found that over a quarter of women (26.7%) reported some form of coercion during childbirth [82]. These women were

more likely to experience 'informal coercion' during an assisted birth or caesarean. They were also more likely to report low satisfaction with the birth and had an increased risk of experiencing postpartum depression.

Consistent with prior research [83–85], how well women felt supported both during and post birth influenced their perceptions of childbirth, reports about their postnatal mood, and descriptions of their newborn baby's physiological and behavioural wellbeing. Mothers were grateful for good emotional support during labour, and professional postnatal support was generally considered helpful by mothers who initially struggled with breastfeeding. However, mothers could at times feel neglected by health professionals, which some presumed to be due to staff shortages or because they appeared to be coping well alone.

Separation from her partner or baby could also colour maternal perceptions of the birth and her newborn baby's behaviour. Flacking and colleagues [86, 87] found separation to be especially pertinent during periods of neonatal intensive care where the basic physical and emotional needs of mother and newborn infant to be together may be supplanted by the infant's need for urgent medical care.

## Postnatal days with baby

Early postnatal events could at times affect maternal wellbeing and mothers' confidence in coping and caring for their baby. Mothers who had more positive birth experiences were more likely to report immediate skin-to-skin contact and successful breastfeeding. We know that breastfeeding initiation and duration can be affected by use of pharmacological pain relief such as pethidine or epidural [72], or by having a caesarean section [88], and many of the mothers in the present study reported struggling with breastfeeding after these experiences. However, receiving breastfeeding support appeared to aid in healing a negative experience, and in turn, encouraged mother-infant bonding also. This could be due in part to increased skin-to-skin contact during breastfeeding.

'Skin-to-skin' contact after childbirth encourages endogenous oxytocin release in mother and infant, enabling the newborn to exhibit instinctive behaviours such as the 'breast crawl' [89]. It has also been found to increase self-efficacy and breastfeeding confidence in primiparous mothers [90]. While one study found these breastfeeding benefits only for the first month [91], sustained skin-to-skin contact may bring multiple benefits, including decreased neonatal crying, more successful breastfeeding [92], and improved self-regulation in the infant [93]. Conversely, separation is experienced as distressing for both mother and newborn [94]. When mothers in our sample felt well-supported and encouraged to have close contact with their baby, and to persevere with breastfeeding after a difficult birth or postnatal experience, this seemed to have a mutually restorative effect, and some mothers felt their unsettled newborn infants became calmer and easier over time [95, 96].

A few mothers who experienced a traumatic birth perceived their baby as difficult, needy or clingy, especially in the early postnatal period. Some mentioned that their own physical and emotional state may have been recognised and reflected by their baby. We know that negative postpartum maternal mood can affect infant temperament and development [97] and may itself be directly influenced by childbirth experience [98]. In turn, it may affect mothers' perceptions of their baby [99]. Furthermore, an unsettled infant could lead to further maternal distress [100]. Negative postpartum mood may also affect mother-infant interactions, resulting in more insecure attachments [101–104]. Subsequently, insecure attachments can affect infant emotional self-regulation and behaviour [28].

Some participants tried to compensate and help to settle their babies after a traumatic birth by 'keeping them close'. Wolke [105] highlights how sensitive maternal caregiving may help

the infant in their 'behavioural organisation'. Maternal sensitivity to infant distress also mediates attachment [106] and may especially benefit infants with negative emotionality [107].

## Limitations

While this study deepens our understanding of mothers' perceptions of how their physical and psychological birth experiences may affect the temperament of their infant, thus adding to the existing body of literature [e.g., 30, 108], it is not without limitations. Although maternal reports are considered to contain some level of accuracy regardless of birth experience [109–111], this sample is too small to enable generalisation of the findings to a wider population. A much larger study would be needed to test the potential pathway from birth, postnatal mood, bonding and attachment to early infant temperament.

Despite concerted efforts to recruit a representative sample across Southwest regions of England and Wales, the study had little sociodemographic diversity. For instance, all participants were British Caucasian, and most had achieved a higher level of education. Furthermore, the majority of mothers in this study (n = 15, 68%) breastfed their infants–a much higher proportion than reported elsewhere (e.g., approximately 44% of mothers with infants aged 6–8 weeks [112, 113]). Therefore, any further study around this topic should aim to include mothers from different cultures and socioeconomic backgrounds to improve the representativeness of the sample.

## Conclusions

Although many mothers did not perceive any explicit connection between their birth experience and their baby's behaviour, others spoke of a potential link between the two. As found in our previous qualitative study of maternity care providers [108], the findings here suggest that mother and infant wellbeing is mutually determined. Whatever the mode of birth, the way mothers experience childbirth and are cared for during and after birth may influence how they perceive and cope with their infants' early behavioural style. While further research is needed on this topic, these findings add to previous literature emphasising the importance of supportive care, including the need to ensure that all women receive personalised, empathic one-to-one care, and consistent emotional as well as physical support during and after childbirth. This may benefit not only mothers but also the present and future wellbeing of their infant.

## Acknowledgments

Thank you to all the mothers who took part and shared their experiences and perceptions.

## Author Contributions

**Conceptualization:** Carmen Power, Claire Williams, Amy Brown.

**Data curation:** Carmen Power.

**Formal analysis:** Carmen Power, Claire Williams, Amy Brown.

**Investigation:** Carmen Power.

**Methodology:** Carmen Power, Claire Williams, Amy Brown.

**Project administration:** Carmen Power, Claire Williams, Amy Brown.

**Supervision:** Claire Williams, Amy Brown.

**Writing – original draft:** Carmen Power.

**Writing – review & editing:** Carmen Power, Claire Williams, Amy Brown.

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
