## [Decision Letter · Decision Letter 0]

14 Dec 2022

PONE-D-22-24651Does a mother’s childbirth experience influence her perceptions of her baby’s behaviour? A qualitative interview studyPLOS ONE

Dear Dr. Williams,

Thank you for submitting your manuscript to PLOS ONE. After careful consideration, we feel that it has merit but does not fully meet PLOS ONE’s publication criteria as it currently stands. Therefore, we invite you to submit a revised version of the manuscript that addresses the points raised during the review process.

We look forward to receiving your revised manuscript.

Kind regards,

Adetayo Olorunlana, Ph.D.

Academic Editor

PLOS ONE

Journal Requirements:

Reviewers' comments:

Reviewer's Responses to Questions

**Comments to the Author**

1. Is the manuscript technically sound, and do the data support the conclusions?

Reviewer #1: Partly

Reviewer #2: Partly

2. Has the statistical analysis been performed appropriately and rigorously? 

Reviewer #1: N/A

Reviewer #2: N/A

3. Have the authors made all data underlying the findings in their manuscript fully available?

Reviewer #1: Yes

Reviewer #2: Yes

4. Is the manuscript presented in an intelligible fashion and written in standard English?

Reviewer #1: Yes

Reviewer #2: Yes

5. Review Comments to the Author

Reviewer #1: Please see uploaded attachment.

I have made extensive comments, but this doesnt detract from the fact that this is a potentially important and valuable paper. I do hope the authors are not put off from resubmitting as a result of the issues I have raised.

Reviewer #2: Introduction: Well written, provides relevant previous research. I think that the justification for carrying out the study could be strengthened more. Especially as there is a lot of rich data reported that doesn't just relate to infant behaviour. You could talk about how important women's experiences of birth are, because of the data that shows the long term impact, therefore you will be adding to this body of research, identifying types of practice that are associated with good care, which has the ability to improve care and change practice. The results focus a lot on feeling prepared, so you could also discuss the importance of good antenatal care and birth education, and birth expectations and experiences research.

Regarding the second part of the study, about birth and infant's behaviour, there is already research that drugs and type of birth can affect baby/breastfeeding https://www.sciencedirect.com/science/article/pii/S0266613815000078?casa_token=v6ULrCszbjEAAAAA:sd86yyjPygL-tgH6o1qumGARn-kA6lEz07cGUOgriONnbz89I3qlljrKI0rQSgVpj1obguy-;
https://www.sciencedirect.com/science/article/pii/S0266613821001972?casa_token=I4NqVIXHftUAAAAA:XU2cIZSco3k3e6tTpBGhs6Vm0NOn2H8033EQGJg2mBsoGRiYn6ozLRrGDlolNJIQa2Yguelp; including your study, so is the point of this part of the study to add to this body of research? It isn't entirely clear what the justification of this part of the study is, considering it would be methodologically better to answer this from a quantitative perspective, especially regarding part 3 (associations), so I think this part of the study needs more justification.

Method: Interview schedule. My main comment is about why you asked "how were you induced?". Were all women who took part induced? Or was this question only asked to women who were induced? Could you make this clearer. The first prompt for Question 3 is also grammatically incorrect. Can you clarify this one please.

I do not agree with the labels of your categories (natural and clinical birth). Due to the importance of language, (https://blogs.bmj.com/bmj/2018/02/08/humanising-birth-does-the-language-we-use-matter/) I would advise changing natural to "physiological"

Please provide examples of easy, mixed or difficult infant behavioural descriptions.

Please provide a more detailed description of the quantitative counts methodology.

Results: As per my previous comment, reconsider the use of the term natural.

Quote from mother 14 about birth plan. Should me be my?

I would not describe foetal monitoring as simple, but I agree with standard. It is standard practice with very little evidence to suggest it improves outcomes, so add this to the discussion.

The maternal caregiving theme is not very strong, and may require more description/examples.

Part 3 - there is a lot of methodology here and I would argue this should be in the method section.

Page 26, first paragraph, there is a double F on the word For.

Discussion: Provide a description of what was positive and negative about women's birth experiences, as this is a large part of your results.

I think your discussion potentially goes too far in trying to work out the relationship between women's birth experiences and infant behaviour, especially when this is a qualitative study and you cannot really draw these conclusions, especially with a sample of only 22 women. The discussion should focus more on women's experiences of birth, and how they thought it impacted their baby. Their experiences were affected by their type of birth and the way they were treated during birth, which adds to a large body of evidence on this already. More focus should be made about the coercion and neglect women experienced, which goes against the initiatives of multiple organisations (White Ribbon Alliance, WHO). Then you should go onto say it is extra important that maternity care is improved because the results suggest that birth experience MAY impact infant behaviour, but more research is needed.

You can also talk about the need for good preparation, good antenatal care, good support. All of which has been found by previous studies, and your study supports.

6. PLOS authors have the option to publish the peer review history of their article (what does this mean?). If published, this will include your full peer review and any attached files.

Reviewer #1: No

Reviewer #2: No

---

## [Author Response · Author response to Decision Letter 0]

16 Mar 2023

Response to Reviewers 

Dear Reviewers

Thank you for your helpful, constructive and supportive comments. For ease of reference, we have captured our responses and accompanying changes in the itemised list below. Please note that cited page numbers correspond to the manuscript file with track changes. 

General comments to both reviewers

We have adjusted wording throughout the paper to be more cautious when talking about mothers’ perceptions of infant behaviour after childbirth. Reference to quantitative methodology has been reviewed and adapted to reflect the largely qualitative approach to the study. Outcomes are no longer categorised and wording has been adjusted in relation to ‘natural’/‘clinical’ births throughout the paper.

Reviewer 1

1. This is an interesting paper, addressing an important area. It is well written, with good logical progression, and is generally comprehensive. However, I do have some comments and questions about some aspects of the text. Fundamentally, the paper seems to be trying to address too many issues at once. I would suggest that the team carefully consider the primary focus and trim the paper down so that that focus is very clear and consistent.

Response: The focus has been narrowed throughout the paper to emphasise mothers’ own perceptions and thoughts about their birth and their infant’s early temperament.

2. I have made a number of comments about the small sample size. This is not a problem if there are sound and saturated themes, and the claims are being made about the nature of these themes. However, in this case, a number of claims of causation/association are made in the data, and these do need larger sample sizes and more representative samples in order to be substantiated. 

Response: Part 3 has been removed from the original results section, and Part 2 has been incorporated within the mothers’ birth stories. Any references to ‘patterns’ or associations in the data have been removed. Data saturation has been explained further (e.g., second to final paragraph, page 6). 

3. Fit between the aims and the methodological approach: The text seems to suggest that there is already evidence of an association between birth experience and women’s perceptions of infant behaviour. Given this existing information, what this study seems to add is a description of what women believe about the nature of their experiences and perceptions of infant temperament. I would be inclined to take out suggestions of empirical cause and effect, as that is not really the epistemological focus of qualitative research. However, exploring whether mothers believe that the two are related, as in the second part of the text below, is appropriate: so I suggest that this is the emphasis throughout, as in the proposed amendments to the aim below:

“Consequently, the aim of the current study was to further explore how mothers’ subjective childbirth experiences might influence and their perceptions and interpretations of their baby’s behaviour, and whether they believed that the two were related.”

This framing encompasses the results reporting both what the women say they believe, and the ‘researchers view’ in which you provide a critique of what they say they believe in the light of how they talk about their perceptions. It may be useful to structure the paper a bit more clearly at these two different levels of analysis (and I would suggest that some of the more quantitative and ‘causative’ aspects of the paper could be deleted to allow for a more in-depth exploration of the nuances between what is described by the participants, and how they interpret what they describe). 

Response: We have further considered the fit between the aims and methodological approach and edited accordingly throughout (e.g., introduction, first paragraph page 4). The focus of the study is now more about mothers’ perceptions of their birth experience and their infant’s behaviour, as well as discussing any perceived or potential links between the two. ‘Causative’ aspects have also been removed, and the relationship between mothers’ descriptions and beliefs is discussed.

4. Reflexivity is an important corner stone of qualitative research, as it enables the reader to understand which lens(es) are applied to the design, data collection, analysis and interpretation of the study, from the perspective of the researcher pre-suppositions before the study started. A paragraph or two describing the standpoints and beliefs of the research team members in relation to the topic of the paper would add considerable to the credibility and rigour of the study. For instance, the assumption in the abstract and in some parts of the text is that childbirth interventions are associated with traumatic birth. While this is true for some women, there is good evidence that the more important factor overall is how women are treated during labour and birth. Reflecting on this may change both the framing of the issue and the solutions proposed… Did the team look for disconfirming data during the analysis? 

Response: A reflexivity section has been added (page 7), and we have also emphasised how it is the way women are treated during labour and birth more than the birth itself which seems to affect how mothers feel about their birth and their baby’s behaviour. ‘Disconfirming’ examples have also been added to the results section. 

5. This is generally well explained, and the approach to thematic analysis is appropriate. However, I wonder why the data were analysed by birth perceptions as a group and then by perceptions of infant behaviour as a group, rather than looking ‘vertically’ across each birth and infant perception story by individual, to illustrate the nuances of the different patterns of experience and perception? As above, I am not sure about the relevance of the quantification of the data where this occurs.

Response: Part 3 has been removed, and Part 2 has been incorporated within the mothers’ birth stories. Quantitative data has been replaced with more in-depth analysis of the mothers’ quotes. 

6. P 8 The statement is made that the analysis applies Lincoln and Guba’s concept of ‘thick description’, with transferability applying to similar contexts. The key reference for thick description should really be Geertz: for example, Geertz, C. (1973). Thick Description: Towards an Interpretative theory of culture. The Interpretation of Cultures, 3-31. This refers to more than just describing the data, and includes a wider analysis of the underlying relations, interactions, semiotics and etc of the context in which the individuals in the research are situated. I’m not sure this was done in the current study?

Response: Thank you for highlighting this. As we predominantly followed Lincoln and Guba’s (1985, 1986) criteria of trustworthiness, the reference to ‘thick description’ has been removed accordingly. Instead, we focus on Lincoln and Guba’s (1985, 1986) criteria of credibility, transferability, dependability, and confirmability.

7. P8 Do the experience categories of ‘positive’ mean that only positive things were said, negative only negative, and mixed everything else? This is a rather reductionist approach in qualitative terms. I’m also not sure if the coding of positive or mixed or negative birth or easy mixed or difficult infant behaviour was done on the basis of one final question about the overall birth experience, or on the researchers’ interpretation of the whole set of responses from each woman. 

Response: As explained previously, reference to categories and coding of data has now been removed. Instead, a more in-depth analysis of the mothers’ quotes – including ‘disconfirming’ data – has been incorporated throughout. 

8. The demographics seem to cover a good range. Are they representative of the population? What about ethnicity? This is covered in the limitations but it might be worth making a note about the sample representativeness at the outset of the results, to remind the reader. 

Response: A additional note about sample representativeness has been included in the first paragraph of the results section on page 7.

9. Data collection: The questions and example prompts given in Table two are appropriate. How long did the interviews last (the range would be useful here)?. 

Response: Approximate interview length has been captured in the Participants section (Page 4). 

10. The following statement is a good indication of the trustworthiness of the analysis. How was it ascertained that no new or meaningful concepts were being generated?: “Data collection stopped when it was judged that no new and meaningful concepts were being ‘generated’ from the data [40].”

Response: This sentence is now justified by the one that follows and further details about data saturation have been included in the second to last paragraph on page 6. 

11. The results make good use of quote material. Table 3: Part one doesn’t look like themes that have emerged from the data – they are demographics. How many had each of these experiences? (to interpret the credibility of the claims made later). Part two is a bit more nuanced: but what about Joy? Delight? Did these not feature? 

Response: The original themes of ‘birth mode’ and ‘birth environment’ have been renamed to reflect the topics as the mothers spoke about them (i.e., how they influenced the mother’s feelings about her birth experience rather than as simple entities in themselves - P10, Table 3). We have also included a short data summary paragraph to establish how the mothers gave birth (top of P9). We have changed the name of the original subtheme ‘fear’ to ‘Emotional states’ in order to include representation of more nuanced emotions, including ‘joy’. Additional quotes have also been added. 

12. Given the current debates in the UK about terminology for types of labour and birth, there may be a risk in dichotomising assumptions about ‘natural’ and ‘clinical’ birth, and of making assumptions about place and type of birth on very small samples. Indeed, the key point seems to more about the fit or gap between expectations and experience than about place or type of birth? Women’s experiences related to different types and places of birth are already well described in various studies – the contribution to knowledge here seems to be more about whether women say that the way they perceived their labour and birth overall has influenced the behaviour of their infants. Indeed, there does seem to be some disconfirming data for the claims made about type of birth (p17) in the account of a ‘natural’ birth that was then associated with distress (although this seemed to be more about the (lack of) postnatal care than the mode of birth): Mother 10, natural hospital birth, baby in Neonatal Intensive Care Unit [NICU] post birth)

Response: Physical and psychological birth experiences, as well as styles of infant behaviour, are no longer categorised into specific groups but instead talked about more generally. The study is now more about whether women say that the way they perceived their labour and birth overall has influenced the behaviour of their infants. Additional disconfirming data has been incorporated throughout. 

13. Table 5 speaks to the topic but the ‘n’ is needed, and ‘birth mode’ or ‘pain relief’ are demographics not themes – ‘Baby mirrors mother’ or ‘maternal caregiving’ are closer to a thematic concepts. 

Response: The terms ‘birth mode’ or ‘pain relief’ have been changed accordingly, reflecting mothers’ personal experiences and perceptions around these birth events (Table 3, P10). Table 5 has been removed as Parts 2 and 3 (original results) have been folded into Part 1. The ‘n’ is given for details of the birth and infant behaviour in the ‘Data summary’ on P9. 

14. N of how many? Words such as ‘often’ are used throughout, or statements such as ‘waterbirths were…’. Does this mean the majority of respondents said these things? How many waterbirths were there? I would suggest that throughout the text should be more cautious. For example: P23 ‘…..often enabled their baby to be calm and settled’ is better as ‘some who reported overall positive births linked this experience with their babies being calmer and more settled’ 

Response: As mentioned above, the ‘n’ is given for details of the birth and infant behaviour in the ‘Data summary’ on P9. We have also adapted more cautious wording throughout (e.g., ‘often’ replaced with ‘sometimes’ or ‘tended to’) and when we refer to possible rather than definitive outcomes of an experience (e.g., 2nd para P27, last para P28).

15. And the opposite with the following quote about forceps and the behaviour of the baby: Conversely, birth complications sometimes led to pain, distress and unsettled baby behaviours. I would suggest this is changed to: ‘some women suggested that specific birth events (such as forceps delivery) might be linked to specific baby behaviours’

Response: Wording has been amended throughout to align more with mothers’ perceptions of birth events and infant behaviour rather than an interpretation of patterns in the data (e.g., 2nd para P27, last para P28).

16. In both cases, in terms of disconfirming data, were there any accounts from women with a birth coded as ‘positive’ or ‘negative’ who said anything different to the statements above?

Response: Yes, and additional disconfirming data has been included to illustrate (e.g., women who had planned interventions who perceived and reported their infant’s behaviour as calm and content). Whilst reference to ‘positive’ or ‘negative’ (coding) have been removed, there is naturally still reference to mothers who had more positive or negative types of experience for one reason or another – often linked to supportive care or conversely, perceived neglect. 

17. P24 I’m not sure why this quote is used with reference to labour pain/pain relief: She’s definitely a more clingy baby than my first. She does not like to be put down, she doesn’t lie down, she’s quite sicky... She won’t settle away from me.” It maybe that a longer section of this quote shows that the mother is associating this behaviour with labour pain/use of pain relief, and, if so, the quote needs to be expanded to show this. 

Response: The quote has been expanded to provide additional context. 

18. P25 It is good to see a counterfactual case here – but is the mother associating the baby’s colic with her birth experience? Or is she just saying that she (the mother) found it a bit harder to cope than she might given her birth experience? “I think I found her quite difficult because I was a little bit shell-shocked after the birth… in the early days… She was really hard to console for no reason other than maybe feeling a bit colicky…” (Mother 8, MLU waterbirth, no medication) 

Response: This quote now follows Mother 8’s birth story: “Some mothers who had a spontaneous physiological birth also found it challenging” and “I don’t think I had a traumatic birth, but… being told nothing was happening and thinking, I can’t cope with it if that’s the case…” (Mother 8, MLU waterbirth, no medication)’. Prior to the above quote about baby’s behaviour, we have inserted: ‘This mother went on to describe her baby as unsettled, although she did not contemplate any connection between the birth and her baby’s behaviour.’

19. P27 As noted above, I would suggest that, at least based on the data from this study, mode of birth/interventions are not part of the psychological experience measure, but that this should be built on the mothers’ perceptions. The sample size in this study is not big enough to be certain that mode of birth/interventions are definitively associated with the outcome, in the absence of other contextual aspects. 

Response: As captured elsewhere, a more qualitative angle has now been adopted. 

20. P29 The following comment is interesting, but the ‘n’ is very small for a statement of association, and the demographics are not necessarily representative. Any statement like this needs to be made very cautiously, and with the imperative that the findings need to be verified with a much larger sample: Notably, no mother who had a spontaneous physiological birth described her baby’s behaviour as difficult, with the majority describing an easy baby. Conversely, almost half of all clinical births were associated with a description of a difficult baby, with less than a quarter describing their baby as easy. 

Response: In line with changes explained above, content has been realigned to focus on mothers’ perceptions of their birth, baby’s behaviour and their thoughts regarding potential links between the two. 

21. Table 9 also interesting, but, again, it should be noted that this is a small and skewed sample, and the assumptions need to be checked with larger samples. It isn’t quite clear what the direction of effect is. Does a difficult baby reify the birth experience as negative, or vice versa? 

Response: No longer applicable as Table 9 has been removed along with this section. 

22. Based on prior studies, the statement below is important, but, based on an n of two, it cannot be transferable from this study. It would be better to frame this as ‘Two women had planned interventions (one CS and one induction) and both reported positive perceptions’. ‘…The remaining 25% of mothers with positive perceptions of their birth (n = 2) had experienced a planned intervention, such as a planned induction or CS’.

Response: As explained previously, the emphasis is now on mothers’ perceptions of their experiences and infant’s behaviour – with the results section modified accordingly.

23. Discussion: The discussion could be shorter. There are also some new findings here. Some of the assumptions seem to be quite far from the actual data, or, at least, don’t seem to take account of possible alternative explanations. This is less likely to happen when the team have previously considered their reflexivity positions, and deliberately searched for disconfirming data. 

For example, in relation to the two comments below: what about the counter-factual? Might the fact that the newborn is perceived to be unsettled be linked to something else other than the birth? Difficulties in feeding, lack of postnatal support, maternal worry about financial security, for instance: meaning that there is no cognitive dissonance in their accounts in relation to infant behaviour and the birth?: 

P30 - After outlining a physically traumatic birth, some mothers proceeded to describe their newborn as unsettled, without attributing their baby's behaviour in any way to the birth. This might be due to maternal guilt, disassociation or denial after a difficult birth experience [52], and therefore, may not have wished to perceive or disclose a potential association between the birth and their baby’s behaviour 

P31 - Consequently, the lack of consistency in maternal responses could be caused by cognitive dissonance stemming from a desire to believe that their baby had experienced a positive start to life regardless of the type of birth they had experienced

Response: The discussion has been streamlined throughout, and the phrase ‘cognitive dissonance’ has been removed. Instead, we have included other ‘possible alternative explanations’ as suggested (see Discussion first paragraph, P26). 

24. P32 Yes, the statement below is correct, the numbers are too small to make any claims – I would suggest deleting this comment, or reframing it as I have suggested: Although the sample numbers are too small to draw any definitive conclusions, the n = waterbirths were generally associated with maternal perceptions of calmer infants, while the n= unsettled forceps birth infants were considered to have residual pain and distress.

Response: We have amended as follows: ‘Although many mothers perceived no connection between their birth experience and their baby’s behaviour, some who reported an easy, straightforward birth, such as a calm and well-supported waterbirth, connected this experience with their babies seeming content settled. In contrast, others perceived that difficult birth events, such as forceps delivery, might be linked to their baby’s unsettled behaviour due to residual pain and distress.’ (2nd para, P27).

25. P32 The concept of Potential Triggers is interesting in the discussion, and P33 - the findings underpinning the statements made below are only represented by one or two quotes (or no direct quotes). The highlighted phrases in the first para indicate where strong claims are made that should be rendered much more nuanced and tentative (and there are similar observations for the other paras selected): 

Birth environment was considered important in relation to maternal and infant wellbeing. Noisy, brightly lit environments increased maternal stress whereas births taking place at home or in a MLU were generally reported as more physically comfortable and psychologically positive experiences. Women who experienced an in-labour transfer from home or MLU to hospital could find it difficult to acclimatise to their new environment, particularly if the transfer involved interventions which raised their stress levels. Unexpected or traumatic circumstances affected maternal reports of infant behaviour. Therefore, the birth environment seemed to have profound psychological as well as physical impacts on the mother and infant.

Midwifery services were often stretched despite the most recent National Institute for Health and Care Excellence (NICE) recommendations of one-to-one care for all labouring women [51]. Inconsistent care and noticeable staff shortages on the postnatal ward were experienced as distressing, with staff shortages on wards known to negatively impact on quality of care [63, 64]. The resulting maternal distress could adversely impact newborn infant behaviour. In contrast, women who had a calm, supportive birth were more likely to attribute their baby’s calm, settled behaviour to the birth environment. 

Response: We have amended the discussion section throughout, including revising wording to ensure claims are more nuanced and tentative. 

26. Do the data actually say this?: P35 c) Health professional authority; Perceived ‘health professional authority’ occurred when women experienced a loss of control. Participants who were not fully informed or felt ‘coerced’ into consenting to unplanned interventions often had regrets around their birth.

Response: These are substantiated by the data, and we have since added additional information/quotes to make this clearer: 

“And then everything I didn’t want to happen happened which was, I didn’t want to have stirrups or anything like that, and the next thing was they took the gas and air away and said I wasn’t pushing hard enough…” (Mother 2, hospital induction) 

“If the doctors decide this is the way they want you to do it it’s very hard not to. If something then happened… I’d never forgive myself.” (Mother 15, hospital induction, forceps)

“They broke my waters – they didn’t ask. Suddenly there was stuff coming out of me. I did not know what was happening … They were telling me, ‘Keep your legs open, keep your legs open’, though my natural instinct was to curl up in foetal position to protect myself.” (Mother 1, overnight hospital induction)

27. There are other statements in the discussion that don’t seem to relate to the findings above. I suggest you check for and amend or remove these throughout. I would suggest as above that the focus of the discussion stays on maternal perceptions of their births and the behaviour of their infants. 

Response: We have now amended the discussion to be more consistent with the data, either by removing ‘claims’ or by adding illustrative quotes. The focus of the discussion now better aligns with maternal perceptions of births and their infant’s behaviour.

28. P40 The limitations section is rather long. Also, in qualitative terms, and given the aim of this paper to understand women’s reported experiences, I don’t think the following is a limitation, since the whole point (as I understand it?) is to capture women’s subjective experiences: However, parents may also be influenced by prenatal expectations, and their emotional involvement naturally precludes them being entirely objective observers of their infant’s temperament. The limitations about the sample are accurate but are not always reflected in the discussion of the meaning and potential transferability of the study findings.

Response: We have shortened, simplified and reworded this section based on your comments. 

29. P39 suggest changing: Whilst this study deepens our understanding of how physical and psychological birth experiences may affect early infant temperament…To: Whilst this study deepens understanding of women’s perceptions of how their physical and psychological birth experiences may affect the temperament of their infant… 

Response: Amended accordingly. 

30. P40 – While it may be true that what the researcher sees as ‘difficult’ is not necessarily in line with the woman’s perception and that women MAY be engaging in self-deception for self-protection, this is an assumption that values the researcher interpretation over what the woman herself says. In fact, the woman may feel no need for self-protection – she may simply be telling it as it is from her perspective. Both points of view need to be discussed here. 

Response: Both points of view are now discussed, with mothers’ perspectives taking precedence throughout. We have also amended conclusions to focus on implications for making maternity care improvements to support mothers better during childbirth and postnatally. 

31. It is important to explain and reference in the discussion how better one-one care and emotional support during childbirth help with women’s perceptions of their birth, their infant’s behaviour, and any associations between these two, if this is to be a key conclusion of the study.

Response: In line with this, we have explained further how better one-one care and emotional support during childbirth (as opposed to neglect) might help with women’s perceptions of their birth and their infant’s behaviour (e.g., final para, P28, reference 75; and 2nd para on P30). 

Reviewer 2

1. Introduction: Well written, provides relevant previous research. I think that the justification for carrying out the study could be strengthened more. Especially as there is a lot of rich data reported that doesn't just relate to infant behaviour. You could talk about how important women's experiences of birth are, because of the data that shows the long-term impact, therefore you will be adding to this body of research, identifying types of practice that are associated with good care, which has the ability to improve care and change practice. The results focus a lot on feeling prepared, so you could also discuss the importance of good antenatal care and birth education, and birth expectations and experiences research.

Response: Further justification has been added to the abstract and introduction sections. We also mention birth expectations and experiences research in the introduction and discuss this more fully in the discussion section as well. Although the mothers were not specifically asked about antenatal preparation, we have referred (where appropriate) to the concept in the discussion (e.g., 3rd para, P29). 

2. Regarding the second part of the study, about birth and infant's behaviour, there is already research that drugs and type of birth can affect baby/breastfeeding https://www.sciencedirect.com/science/article/pii/S0266613815000078?casa_token=v6ULrCszbjEAAAAA:sd86yyjPygL-tgH6o1qumGARn-kA6lEz07cGUOgriONnbz89I3qlljrKI0rQSgVpj1obguy-;
https://www.sciencedirect.com/science/article/pii/S0266613821001972?casa_token=I4NqVIXHftUAAAAA:XU2cIZSco3k3e6tTpBGhs6Vm0NOn2H8033EQGJg2mBsoGRiYn6ozLRrGDlolNJIQa2Yguelp; including your study, so is the point of this part of the study to add to this body of research? It isn't entirely clear what the justification of this part of the study is, considering it would be methodologically better to answer this from a quantitative perspective, especially regarding part 3 (associations), so I think this part of the study needs more justification.

Response: Thank you for these additional references (now references 88 and 89), which we have included in the discussion section under Postnatal days with baby (3rd para, P30). Our study adds mothers’ qualitative experiences (e.g., of medication such as pethidine during labour and breastfeeding support postnatally, and how these factors may have influenced infant behaviour) to the body of existing evidence. To clarify that it is a qualitative study, we have now removed parts 2-3 of the original results section (see response to Reviewer 1 comments). Instead, we interweave the perceived infant behavioural outcomes with the mothers’ birth experience stories.

3. Method: Interview schedule. My main comment is about why you asked "how were you induced?". Were all women who took part induced? Or was this question only asked to women who were induced? Could you make this clearer.

Response: In relation to the ‘Semi-Structured Interview Schedule for Mothers with example prompts’, we have added some additional text to clarify that these are examples of prompts rather than set questions. For example, ‘If you were induced, how were you induced and how did that feel for you?’, 

4. The first prompt for Question 3 is also grammatically incorrect. Can you clarify this one please?

Response: Thank you. This has now been corrected. 

5. I do not agree with the labels of your categories (natural and clinical birth). Due to the importance of language, (https://blogs.bmj.com/bmj/2018/02/08/humanising-birth-does-the-language-we-use-matter/) I would advise changing natural to "physiological".

Response: Considering all reviewer comments, the results section has been amended and no longer categorises the main variables of childbirth and infant behaviour. We agree with the importance of ‘humanising’ childbirth experiences, including the significance of language used. Consequently, we have amended natural birth to ‘spontaneous’ and/or ‘physiological’ to keep in line with current recommendations and NICE guidelines.

6. Please provide examples of easy, mixed or difficult infant behavioural descriptions.

Response: Whilst we previously gave examples of mothers’ behavioural descriptions of easy, mixed or difficult in Table 7, this is no longer applicable. The paper has been restructured and no longer categorises birth experiences or infant behaviour. Instead, it now focusses on mothers’ qualitative descriptions of their birth and infant behaviour, and the implications of this for improving the quality of maternity and perinatal care. 

7. Please provide a more detailed description of the quantitative counts methodology. 

Response: Rather than attempting to quantify such a small sample, the content analysis was conducted to summarise the data and further clarify the picture it appeared to be presenting. However, we have now removed all quantitative methodology from the paper, instead focussing the paper on mothers’ descriptions of their births and their baby’s behaviour, and where mothers sometimes felt there was a possible connection between the two. 

8. Results: As per my previous comment, reconsider the use of the term natural.

Response: We have changed all references to natural birth to ‘spontaneous’ and/or ‘physiological’ to keep in line with current recommendations and NICE guidelines.

9. Quote from mother 14 about birth plan. Should me be my?

Response: the quote is verbatim, where Mother 14 said ‘me birth plan’. 

10. I would not describe foetal monitoring as simple, but I agree with standard. It is standard practice with very little evidence to suggest it improves outcomes, so add this to the discussion.

Response: We have changed ‘simple’ to ‘standard’ to describe electronic foetal monitoring. We agree with the lack of evidence and have added this to the discussion (last paragraph, P29). 

11. The maternal caregiving theme is not very strong and may require more description/examples.

Response: We have added extra text and examples to the maternal caregiving theme. 

12. Part 3 - there is a lot of methodology here and I would argue this should be in the method section.

Response: As mentioned above, we have reconsidered the different methods applied to the data and narrowed the focus of the study to be more in keeping with its qualitative aims and objectives. Therefore, Part 3 has been removed. 

13. Page 26, first paragraph, there is a double F on the word For.

Response: This has been corrected, thank you. 

14. Discussion: Provide a description of what was positive and negative about women's birth experiences, as this is a large part of your results.

Response: Although they are no longer categorised, we now describe more fully how mothers can have more positive or negative experiences of childbirth and the care they received in the final paragraph on page 28. 

15. I think your discussion potentially goes too far in trying to work out the relationship between women's birth experiences and infant behaviour, especially when this is a qualitative study and you cannot really draw these conclusions, especially with a sample of only 22 women. The discussion should focus more on women's experiences of birth, and how they thought it impacted their baby. Their experiences were affected by their type of birth and the way they were treated during birth, which adds to a large body of evidence on this already. More focus should be made about the coercion and neglect women experienced, which goes against the initiatives of multiple organisations (White Ribbon Alliance, WHO). Then you should go onto say it is extra important that maternity care is improved because the results suggest that birth experience MAY impact infant behaviour, but more research is needed. You can also talk about the need for good preparation, good antenatal care, good support. All of which has been found by previous studies, and your study supports. 

Response: We amended the focus from trying to draw conclusions about the relationship between women's birth experiences and infant behaviour, to discussing how women experienced their labour, birth and postnatal days and how they felt this may have affected their baby’s wellbeing and behaviour post birth. As suggested, we have also added parts to the discussion under ‘psychological birth experience’ about the coercion and neglect experienced by some women - in direct opposition to current recommendations and intrapartum guidelines by NICE (2017) and the WHO (2018), both of which inform the White Ribbon Alliance (e.g., see bottom P28, top P29). As mentioned above, the importance of good preparation and antenatal care is also now also included.

---

## [Editor Report · Decision Letter 1]

27 Mar 2023

Does a mother’s childbirth experience influence her perceptions of her baby’s behaviour? A qualitative interview study

PONE-D-22-24651R1

Dear Dr. Williams,

We’re pleased to inform you that your manuscript has been judged scientifically suitable for publication and will be formally accepted for publication once it meets all outstanding technical requirements.

Kind regards,

Adetayo Olorunlana, Ph.D.

Academic Editor

PLOS ONE
---

## [Editor Report · Acceptance letter]

28 Mar 2023

PONE-D-22-24651R1 

Does a mother’s childbirth experience influence her perceptions of her baby’s behaviour? A qualitative interview study 

Dear Dr. Williams:

I'm pleased to inform you that your manuscript has been deemed suitable for publication in PLOS ONE. Congratulations! Your manuscript is now with our production department. 

Kind regards, 

on behalf of

Associate Professor Adetayo Olorunlana 

Academic Editor

PLOS ONE